# Microbiota Succession and Chemical Composition Involved in Lactic Acid Bacteria-Fermented Pickles

**Xiangna Lin [1], Shalkyt Bakyrbay [1], Lingxiao Liu [2], Xiaojuan Tang [1] and Yunguo Liu [1,*]**

[1] College of Life Sciences, Linyi University, Linyi 276000, China; sevenlinlin@163.com (X.L.); s1220813177@163.com (S.B.); tangxiaojuan@lyu.edu.cn (X.T.)
[2] Linyi Academy of Agricultural Sciences, Linyi 276012, China
* Correspondence: yguoliu@163.com; Tel.: +86-539-7258711

**Abstract:** Pickles are a type of traditional fermented vegetables in China. To ascertain the effect of different lactic acid bacteria on pickles, the chemical composition characteristics, flavor substances, and bacterial diversity of the pickles fermented by natural bacteria, *Lactobacillus plantarum* R5, *Lactobacillus pentosus* R8, and *L. plantarum* R5 plus *L. pentosus* R8 were investigated in this study. The results showed that *Lactobacillus* enhanced the decrease in pH, increase in total acid content, degradation of nitrite, and production of organic acid (lactic acid and malic acid) of fermented pickles. A total of 80 flavors were detected in the pickles fermented for 14 days, and esters in pickles fermented by *Lactobacillus* were more plentiful. Firmicutes emerged as the predominant microbial phyla. Amongst these, the commonly encountered microorganisms were *Lactobacillus*, *unclassified Enterobacteriaceae*, *Pantoea*, and *Weissella*. The multivariate statistical analysis further showed that *Lactobacillus* had a strong negative correlation with pH and a strong positive correlation with malic acid and lactic acid, and the microorganisms in pickles could acclimate to the changing fermentation environment. The insights gained from this study may be of assistance to us in obtaining new insights into the microbiota succession and chemical compounds involved in the pickles fermented by *Lactobacillus*.

**Keywords:** pickles; *Lactobacillus*; chemical composition; flavor; bacterial diversity; correlation analysis

## 1. Introduction

Pickles are a type of traditional fermented vegetable in China, and have been deemed to date back to more than 3000 years [1,2]. Pickling is an ancient gastronomic craft (before 2400 BCE) of preserving food in brine and/or vinegar [3,4]. Traditional Chinese pickles are usually made by the natural fermentation of fresh vegetables such as cabbages [2], cucumbers [5], onions, celery, carrots [6], or beans, mixed with condiments such as salt, sugar, garlic, ginger, red pepper, and pepper. Pickles are a popular food because of their unique flavor and potential benefits for health in China. According to statistics, the Chinese pickle market continues to grow. The market size increased from 26.028 billion RMB in 2009 to 52.10 billion RMB in 2017, with a compound annual growth rate of 9.06%. By 2018, the Chinese pickle market size reached 55.44 billion RMB, with a year-on-year growth of 6.4%. The Chinese pickle market size reached 68.12 billion RMB in 2022, and the market size will reach 98.73 billion RMB by 2028.

Traditionally fermented pickles are usually homemade products obtained through spontaneous fermentation, but nowadays, solutions to the problems of quality, safety, and mass production of pickles are urgently needed [7]. To address the above problems, research on the control of raw materials, microbial ecosystems, and fermentation processes has been carried out [8]. *Lactobacillus* species are generally recognized as safe and edible bacteria, possessing multiple beneficial functions such as anti-oxidation [9], cholesterol reduction [10], regulating intestinal flora [11], blood glucose reduction [12], and anti-depression [13]. Thus, *Lactobacillus* strains have been considered applicable starter cultures

for their use in fermented pickles. Pickles fermented by *Lactobacillus* have a distinctive flavor and positive health effects [14]. With the development of technology, nitrites have been found in pickles, which are harmful to human health, but *Lactobacillus* have been proven capable of reducing nitrites [15–17]. During traditional fermentation, various compounds are produced, degraded, or converted to other compounds, influencing the quality of pickles. The changes derived not only from the vegetables themselves but also from microbial changes. When pickles are fermented by exogenous *Lactobacillus*, the fermentation process is expedited, and lactic acid is produced, thus lowering the pH value and increasing total acid content. Therefore, the composition of organic acid as an essential element influences the quality of pickles. As mentioned, the quality still relies on a delicate microbial balance [18]. Furthermore, increasingly more attention is being paid to the flavor analysis of various fermented foods and drinks; esters [19], aldehydes [20], and ketones have been found in fermented fruit juices and fermented fish. However, to our knowledge, the interaction between the microbial community and chemical compounds (or flavor) is yet unclear.

Based on the background, cabbages fermented by *Lactobacillus* were studied to identify the effect of *Lactobacillus* on the quality of pickles. The cabbages used in this study are one of the major vegetables cultivated in Linyi City, Shandong Province, China, and lactic acid bacteria were isolated and screened from traditional homemade pickles. The chemical characteristics, flavor composition and microbiological characteristics were investigated during the fermentation. Moreover, the relationship between the microbial community and chemical compounds in pickles was analyzed to clarify the fermentation mechanism. The results of this study can provide theoretical support for the industrial production of *Lactobacillus*-fermented foods and could provide further insight into the fermentation mechanism of vegetables.

## 2. Materials and Methods

### 2.1. Bacterial Strain, Medium and Culture Conditions

*Lactobacillus plantarum* R5 and *Lactobacillus pentosus* R8 were isolated from naturally fermented products of Linyi, Shandong Province. 16S rDNA was used to confirm the identity and was examined periodically under the microscope after Gram staining to ensure their purity. They were propagated in MRS broth at 37 °C for 16 h anaerobically after being serially transferred at least three times. The cultured bacteria cells were centrifuged at $8000 \times g$ for 10 min and washed twice using sterile saline solution. The bacterial pellets were inoculated in the pickle at a $6.0 \times 10^6$ CFU/mL initial cell density.

The propagated strain was preserved in the MRS medium supplemented with 40% glycerol at $-20$ °C.

### 2.2. Preparation of Fermented Pickles and Sampling

Fresh Chinese cabbage was purchased from local supermarkets in Linyi, Shandong Province, China. The Chinese cabbages were washed using boiled water and then drained for 2 h before being placed into a sterilized jar (2 L). The 50% cabbage, 3% garlic, 2% ginger, 4% red pepper, 4% sugar, 1% pepper, and 4% salt (sodium chloride) solutions were mixed uniformly with or without lactic acid bacteria. Finally, these materials were placed in the jar and stored at 25 °C for 28 days. Four groups of the fermentation experiments of pickles were prepared: Group 1: naturally fermented pickles (N group) containing 0% lactic acid bacteria; Group 2: pickles fermented by 1% *Lactobacillus plantarum* R5 (R5 group); Group 3: pickles fermented by 1% *Lactobacillus pentosus* R8 (R8 group); and Group 4: pickles fermented by multi-strain including 0.5% *Lactobacillus plantarum* R5 and 0.5% *Lactobacillus pentosus* R8 (M group). The samples of the pickle brine and cabbage leaves (1:1) were collected by sterile operation at 1, 7, 14, 21, and 28 day during the fermentation period. They were stored in sterile tubes at $-20$ °C for analysis after homogenization and centrifugation.

### 2.3. Determination of pH, the Content of Total Acid, Reducing Sugar, Nitrite and Organic Acids

The pH values of samples were measured with a pH meter (PHS–3C, Fangzhou Technology, Chengdu, China) and the pH meter was calibrated using standard buffer solutions of pH 4.00, 6.86, and 9.18.

The content of the total acid in pickles was measured according to the Chinese National Food Standard (GB12456-2021).

The content of the reducing sugar in pickles was analyzed according to the Chinese National Food Standard (GB500.7-2016).

Hydrochloride naphthodiamide was used to determine the content of nitrite per kilogram according to the Chinese National Food Standard (GB 5009.33–2016).

The organic acid in the pickle was separated on a Kromasil C18 reversed-phase column (250 mm × 4.6 mm, 5 μm) at 30 °C. Isocratic elution with the mobile phase composed of 0.1% phosphoric acid solution and methanol (98:2, $v/v$) was performed for 10 min. The organic acids were identified and quantified through their comparative retention times and peak areas according to standards.

All experiments were repeated three times.

### 2.4. Analysis of Volatile Organic Compounds (VOCs) via GC-IMS

The VOCs of the pickle (14 days) were determined using the GC-IMS instrument (FlavorSpec) from Gesellschaft für Analytische Sensorysteme mbH (G.A.S., Dortmund, Germany) and modified according to previous research. In brief, 2 g of each pickle was placed in a 20 mL headspace bottle and incubated at 40 °C for 15 min. Subsequently, a 500 μL sample in the headspace bottle was obtained with an 85 °C syringe and automatically injected by the syringe into the GC-IMS equipment. Nitrogen with a purity of 99.999% was used as the carrier gas, and the VOCs were separated by a through column (MXT-5, 15 m × 0.53 mm) at 45 °C. The gas flow in the GC was as follows: 2 mL/min for 2 min, 100 mL/min for 20 min, 150 mL/min for 30 min, and then the gas flow stopped.

### 2.5. DNA Extraction and PCR Amplification of 16S rRNA Sequence in the Pickles

Total genomic DNAs from pickle samples were extracted using a Mag-Bind Soil DNA Kit (Omega, Norcross, GA, USA) according to the manufacturer's instructions. The DNAs were diluted to 1 ng/μL using sterile water. Then, 16S rRNA genes were amplified using the following universal primer: forward primer 341F (5′- CCTACGGGNGGCWGCAG-3′) and reverse primer 805R (5′-GACTACHVGGGTATCTAATCC-3′) with the barcode and PCR Master Mix (YEASEN, China). Sequencing was performed on an Illumina HiSeq2500 platform (Sangon, Shanghai, China) based on a standard protocol from the manufacturer.

### 2.6. Sensory Analysis

Sensory analysis was carried out according to the Chinese National Food Standard (GB16291.1-2012) by 10 trained panelists. The pickle samples (10 g from each type of pickle) at the different fermentation stages were used by panelists who were required to evaluate the pickles' fermented aroma, sourness, spiciness, and saltiness. In this study, a score of 9 represented the best quality. The lower the score, the worse the quality.

### 2.7. Statistical Analysis

Three repetitions for the fermentation of pickles in each container were carried out. All data were analyzed using SPSS Statistics 22.0 software (SPSS 22.0, Chicago, IL, USA); $p$-values less than 0.05 were considered statistically significant.

## 3. Results
### 3.1. The pH and the Content of Total Acid and Organic Acid in the Pickles
3.1.1. The pH and the Content of Total Acid

The pH values of the pickle during the entire fermentation process are shown in Figure 1A. The pH decreased with the prolongation of fermentation time. The pH of the R5

group dropped at the fastest rate, followed by the R8 group. Meanwhile, the rate of pH decrease in the M group was significantly faster than that of the N group at the early stage of fermentation. As shown in Figure 1A, pH in different treatments decreased rapidly from 1 to 7 day, decreased slowly from 7 to 14 day, and stabilized from 21 to 28 day. At the end of fermentation, the pH of the D group was higher than other groups treated with lactic acid bacteria.

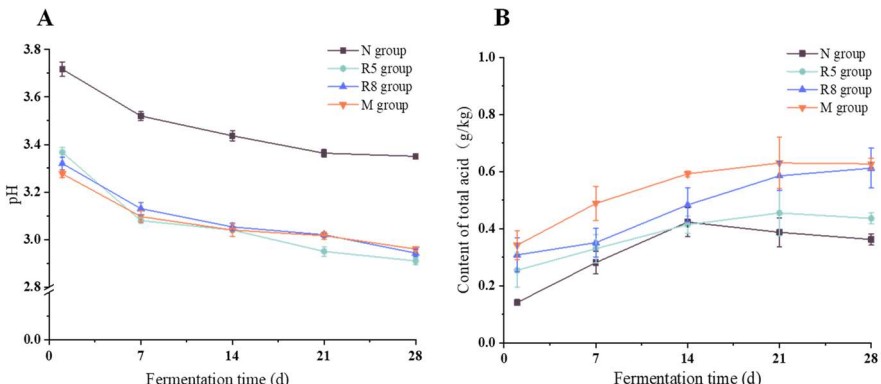

**Figure 1.** Changes in pH and total acid content in pickles during fermentation: (**A**) pH; (**B**) total acid content.

The contents of total acid in pickles during fermentation were increased to different degrees in four groups until they reached a steady state (Figure 1B). The total acid in the initial stage increased faster than that in the middle stage, and four groups showed the same trend. The contents in the N group and R5 group were lower than those of the M group and R8 group.

3.1.2. The Organic Acid in the Pickles

Figure 2 shows the content of organic acid, including lactic acid, acetic acid, malic acid, succinic acid, oxalic acid, tartaric acid, and citric acid, among which the content of lactic acid is the highest, especially in the R8 group (8411.13 μg/g) at 14 day. In the group under natural fermentation, the concentration of lactic acid was lowest from 1 to 28 day compared to other groups. The content of acetic acid in the N group was highest, and its concentration was up to 3534.06 μg/g at 28 day. The content of acetic acid was relatively stable in the different groups from 1 to 28 day. The content of malic acid in different groups gradually increased with time, and in the R5 group, it was the highest at 28 day up to 1752.32 μg/g. The content of succinic acid was highest in the N group.

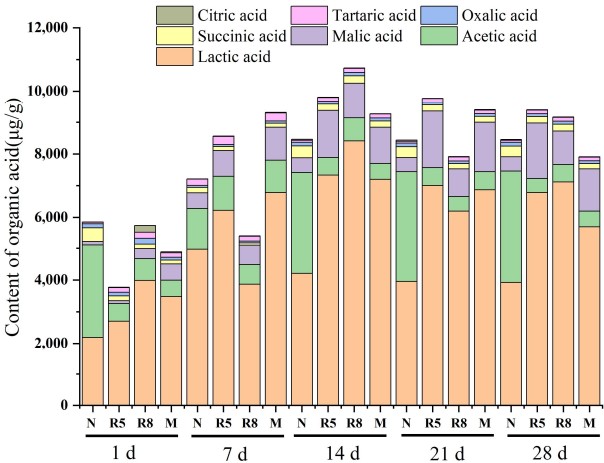

**Figure 2.** Changes in organic acid content in pickles during fermentation.

The oxalic acid content gradually decreased in different treatments, which showed the opposite trend to malic acid. In addition, the content of tartaric acid in the R5 and M groups showed a decreasing trend after a slight increase and reached the highest at 7 day (249.53 μg/g and 251.34 μg/g, respectively). The content of citric acid was the lowest among the measured organic acids. The content of citric acid in the R8 group was 214.82 μg/g on the first day of fermentation and remained stable in the remaining fermentation time.

### 3.2. Content of Reducing Sugar in the Pickles

As shown in Figure 3, the content of reducing sugar in all groups decreased along with the fermentation time. The reducing sugar decreased smoothly and steadily at the initial time and decreased rapidly from 7 day to the end of fermentation.

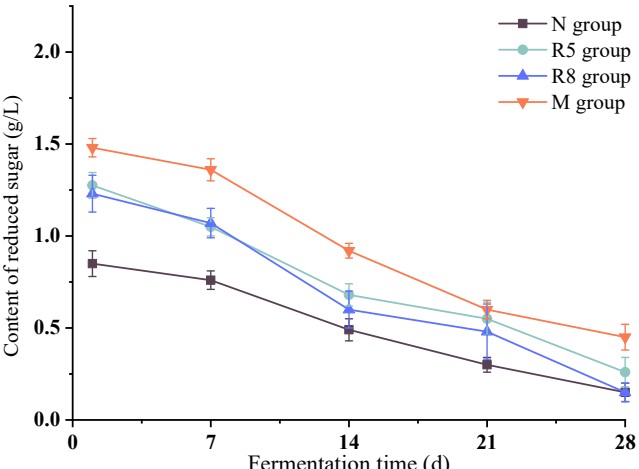

**Figure 3.** Changes in reduced sugar content in pickles during fermentation.

### 3.3. Nitrite Concentration in the Pickles

Nitrite concentration in the pickle went up first, but then it went down (Figure 4). It peaked in different groups at different times. The nitrite concentration in the N group reached the highest (11.92 mg/kg) at 7 day, but in the other groups, it reached the peak at 14 day, and the nitrite concentration was 10.72 mg/kg in the R5 group, 9.07 mg/kg in the R8 group, 9.82 mg/kg in the M group. Over time, the nitrite content gradually decreased, and it decreased to below 2.5 mg/kg at 28 day in all groups.

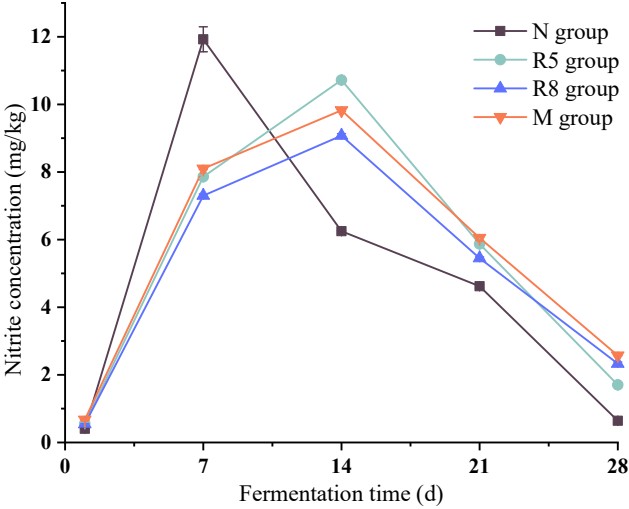

**Figure 4.** Changes in nitrite concentration in pickles during fermentation.

### 3.4. VOCs Analysis in the Pickles

GC-IMS was used to measure the flavor compounds of pickles fermented for 14 days. The topographical plots and fingerprint comparison are shown in Figure 5. A total of 80 flavors were detected, including 16 esters, 7 alcohols, 10 disulfide compounds, 4 aldehydes, 4 ketones, 8 alkenes, 2 acids, 1 phenolic compound, and 1 ether compound. There was less acid in pickles, such as isovaleric acid and 2-methyl propionic acid. The compounds that changed under the influence of the *Lactobacillus* strain were determined by comparing the spot intensity.

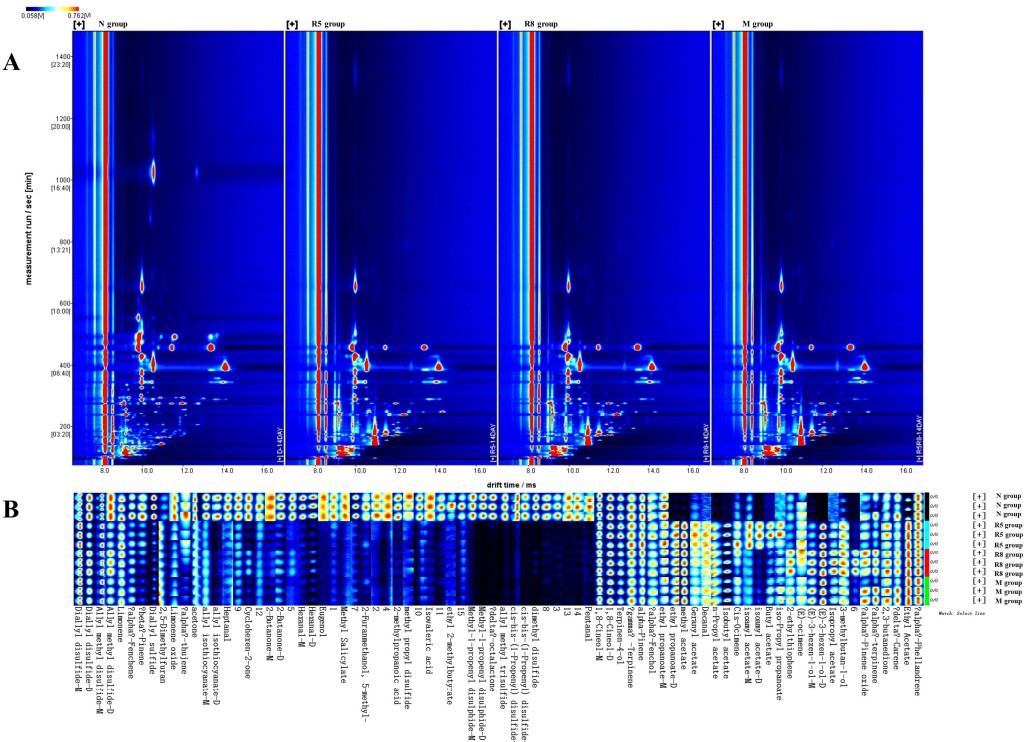

**Figure 5.** Topographical plots corresponding to GC-IMS signals detected in pickles with different starters fermented for 14 days. (**A**) Fingerprint comparison of flavor compounds in pickles with different starters fermented for 14 days determined by GC-IMS. (**B**) The darker the spot is, the larger the quantity of volatile compounds. Each row represents all the signal peaks selected in a sample. Each column represents the signal peak of the same flavor compounds in different samples. The GC-IMS was carried out in triplicate.

As shown in Figure 6, the types and concentrations of VOCs in the samples inoculated with *Lactobacillus* were different from those in the naturally fermented pickles. The content of substances in the naturally fermented pickles was obviously higher than that in other types of pickles, such as delta-octalactone, dimethyl disulfide, allyl methyl trisulfide, isovaleric acid, hexanal, heptanal, 2-methyl propionic acid, and 2-butanone. As measured, the content of sulfur compounds and most terpenoids was the highest in the naturally fermented samples. The composition of VOCs in the pickles fermented by a single strain R5 or R8 is highly similar to that in the pickles fermented by mixed strains of R5 and R8 (Figures 5 and 6A). Esters in the pickles fermented by *Lactobacillus* were more plentiful than those in the naturally fermented pickles, such as ethyl acetate, methyl acetate, ethyl propionate, n-propyl acetate, isobutyl acetate, isoamyl acetate, butyl acetate, iso-propyl propionate, and geraniyl acetate. The correlation among the major component is shown in Figure 6B. The sulfur compounds had a strong positive correlation with ketone and a strong negative correlation with esters.

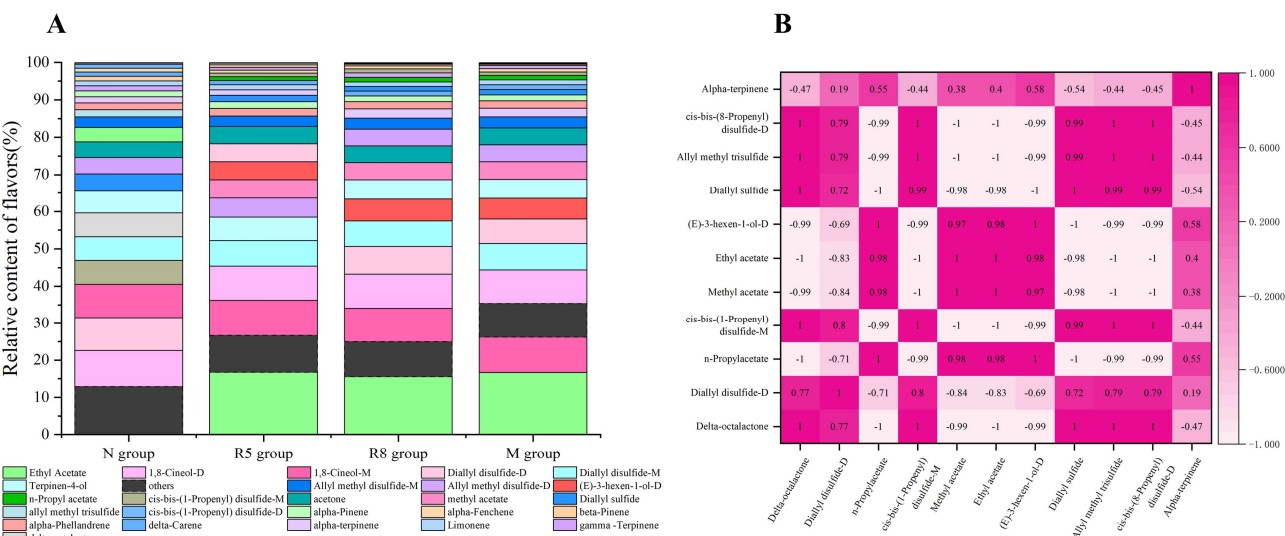

**Figure 6.** The relative content of flavors in pickles. (**A**) Correlation matrix of Pearson's rank correlation among the major component. (**B**) Pearson's rank correlation coefficient ranges from 1.0 to −1.0, corresponding from strongly positive to strongly negative correlation. Only content with an abundance >0.1% in at least one sample is shown in (**A**).

### 3.4.1. Sequencing Information Statistics and α Diversity Analysis

A total of 1687,985 high-quality sequences in bacteria were obtained from 20 samples through Illumina MiSeq sequencing analysis with an average effective sequence length of 401.5–432 bp.

The total number of operational taxonomic units (OTUs) detected at a 97% sequence identity was high in all samples, with up to 140 OTUs for bacteria.

The results showed the following: the larger the Chao and ACE, the higher the community richness; the smaller the Simpson index, the larger the Shannon index, and the higher the community diversity. As shown in Table A1, Chao and ACE in different treatments were increased and then decreased except for multi-strain fermented pickles, meaning the community richness reached the highest when fermented for 14 or 21 days with different treatments. The sequence of the community richness was M group > R5 group > N group > R8 group. The community diversity showed a decreasing trend in different treatments along with the fermentation time, and the community diversity of the naturally fermented pickles was higher than that of other types of pickles. The sequence of the community diversity was N group > R8 group > R5 group > M group.

### 3.4.2. Bacterial Communities of Pickle Samples

An overview of the pickles' microbial community structures at the phylum and genus levels is shown in Figure 7A,B. A total of 20 traditional fermented samples collected under different treatments had different bacterial community compositions at different times. All the pickle samples contained bacteria from the phylum Firmicutes (56.80–97.66%), Proteobacteria (1.91–35.34%), Cyanobacteria (0.05–6.37%), and others (species with an abundance of <1% in all samples). In general, Firmicutes and Proteobacteria were the dominant phyla of bacteria in pickles. Moreover, Cyanobacteria content in the samples collected from naturally fermented pickles was higher than that in the other samples. Bacterial communities at the genus level are shown in Figure 7B; 13 genera constituted 0.1% of the 20 samples.

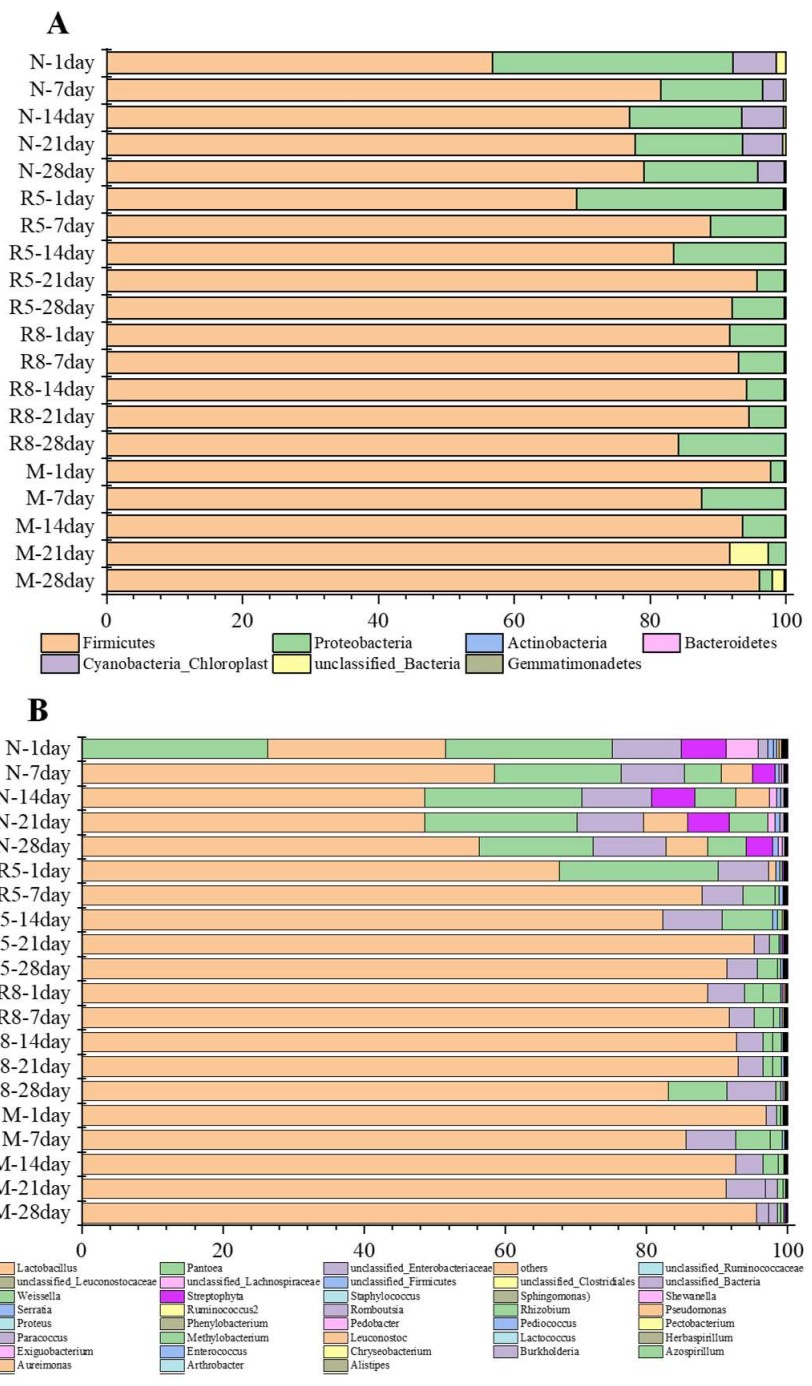

**Figure 7.** The distribution of bacteria in pickles at the phylum (**A**) and genus (**B**) levels. The relative abundance was estimated from 16S rRNA gene sequences.

A heatmap of bacterial diversity at the genus level in different pickles is shown in Figure 8A, suggesting that the dominant genera were similar in the *Lactobacillus*-fermented pickles. Among the identified genera, *Lactobacillus* (0.36–96.96) dominated in all samples except the samples from the N group fermented for one day. The genera with a relatively high abundance were *unclassified_Enterobacteriaceae* (1.22–10.44%), *Pantoea* (0.40–23.55%), *Weissella* (0.21–26.32), *Leuconostoc* (0.02–25.25%), *Streptophyta* (0.05–6.36%), *Exiguobacterium* (0.01–4.48%), *unclassified_Bacteria* (0.02–5.60%), and others (species with an abundance of <1% in all samples).

To determine the correlation between the genera, the abundance of genera >1% was analyzed through Spearman's rank method, as shown in the correlation matrix in Figure 8B. The result indicated that *Lactobacillus* had a strong negative correlation with *unclassified Enterobacteriaceae*, *Pantoea*, and *Leuconostoc*. *unclassified Enterobacteriaceae* had a strong positive correlation with *Pantoea* and *Leuconostoc*. *Weissella* is positively correlated with *Exiguobacterium* and *unclassified Enterobacteriaceae*.

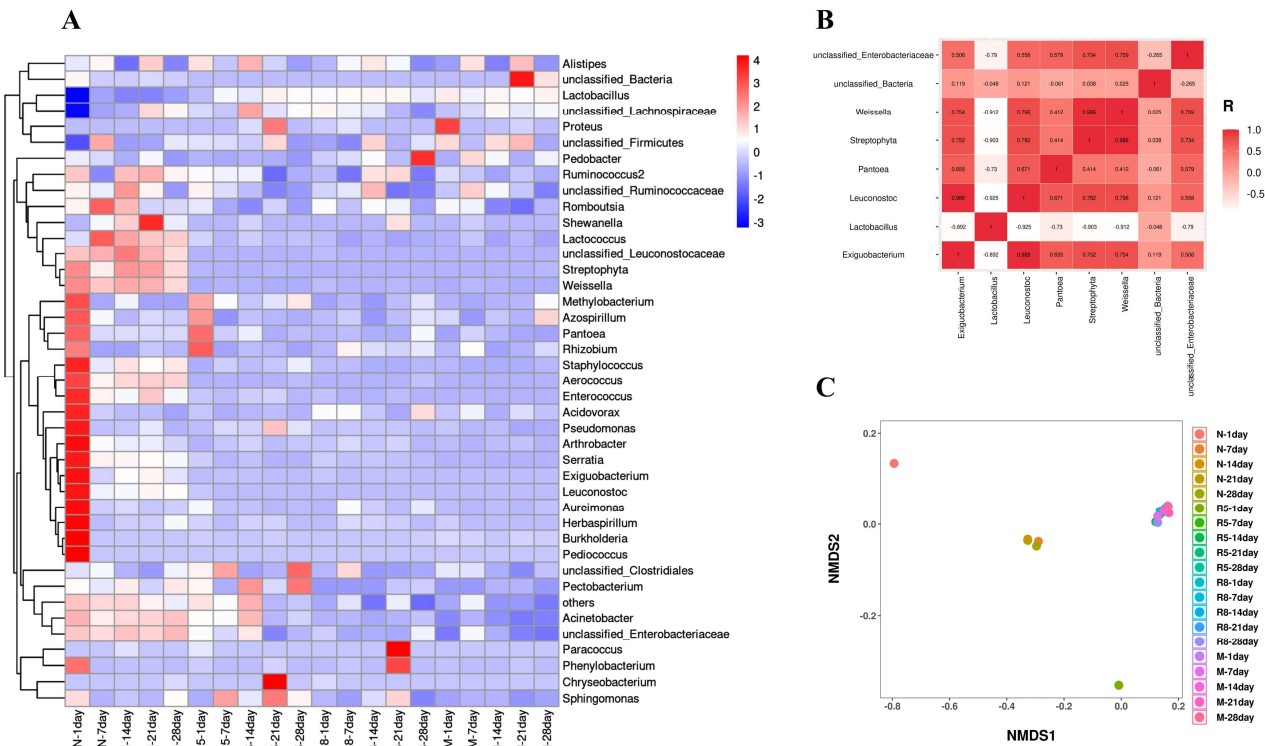

**Figure 8.** Heatmap of bacterial diversity at the genus level in different groups. (**A**) Correlation matrix of the Spearman's rank correlation among the major genera. (**B**) The Spearman's rank correlation coefficient ranges from 1.0 to −1.0, corresponding from strongly positive correlation to strongly negative correlation. NMDS analysis of bacterial community composition in pickle samples (**C**). Only the genus with an abundance >1% in at least one sample are shown here.

A non-metric multidimensional scaling (NMDS) was carried out to investigate the correlation between *Lactobacillus* and the microbiota in the pickles. The most critical categorical variables, including *Lactobacillus*, whether or not to be inoculated, and species, were considered. The bacterial community of the pickles is shown in Figure 8C. The bacterial community in the naturally fermented pickles was different from that in other pickles, especially N-1 samples, but the bacterial communities in the pickles fermented by R5, R8, and R5 plus R8 were similar.

### 3.4.3. Correlation Analysis of Microbial Community and Chemical Composition

To determine the correlation between the genera and chemical index of pickles, an abundance of genera >1% in all the treatments and chemical data, including pH, total acid, nitrite, reducing sugar, and organic acid, were analyzed through Pearson's rank method and shown in the correlation matrix and network correlation in Figure 9A. The result indicated that *Lactobacillus* had a strong negative correlation with pH and a strong positive correlation with malic acid and lactic acid. The pH in pickles had a strong positive correlation with *Exiguobacterium*, *Leuconostoc*, *Pantoea*, *Streptophyta*, *Weissella* and *unclassified Enterobacteriaceae*. Malic acid had a strong negative correlation with *Streptophyta*, *Weissella*

and *unclassified_Enterobacteriaceae*. Lactic acid was similar to malic acid and had a strong negative correlation with *Streptophyta* and *Weissella*.

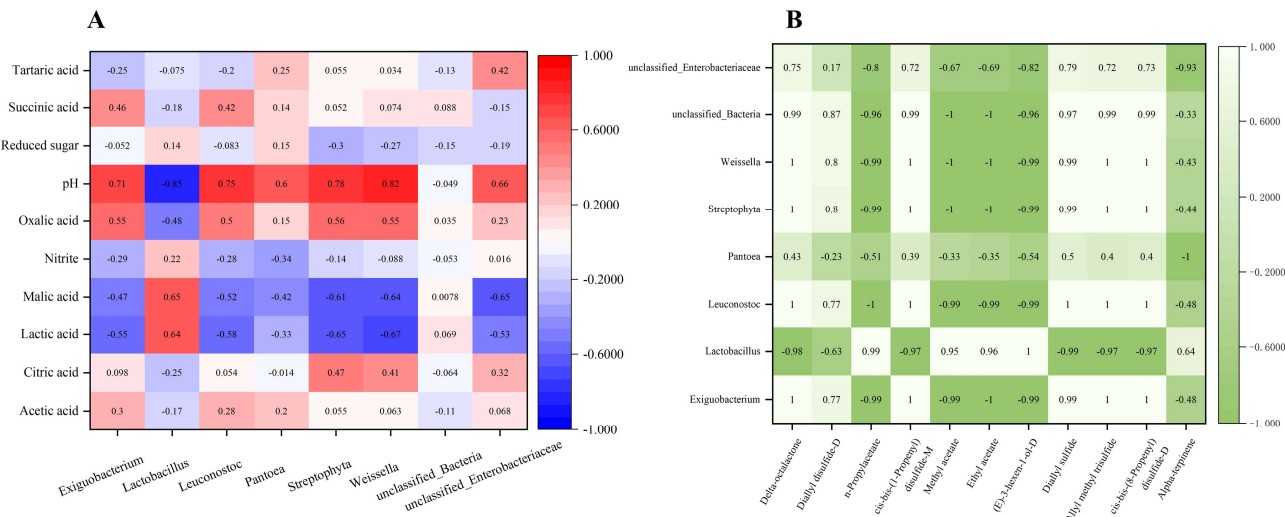

**Figure 9.** The correlation heatmap between bacteria genera and chemical compounds (**A**); between bacteria genera and VOCs (**B**) of the fermented pickles.

### 3.4.4. Correlation Analysis of Microbial Community and VOCs

The correlation between bacteria genera and VOCs of the fermented pickles is shown in Figure 9B. The VOCs with a relative content >1% were used for analysis. The results indicated that *Lactobacillus* had a strong negative correlation with delta-octalactone, cis-bis-(1-propenyl) disulfide-M, diallyl sulfide, allyl methyl trisulfide, and cis-bis-(1-propenyl) disulfide-D, but had a strong positive correlation with n-propy lacetate, methyl acetate, ethyl acetate, and alpha-terpinene. The sulfur compounds had a strong positive correlation with *unclassified_bacteria*, *Weissela*, *Streptophyta*, *Leuconostoc*, and *Exiguobacterium*. The esters had a strong positive correlation with Lactobacillus. The alpha-terpinene had a strong negative correlation with *unclassified_Enterobacteriaceae* and *Pantoea*.

### 3.5. Sensory Evaluation

The pickles produced by spontaneous fermentation and inoculated fermentation for different days were subjected to sensory analysis. Figure 10 shows the sensory profiles of fermented aroma, sourness, spiciness, and saltiness. In different groups, the sensory scores decreased after increasing during fermentation, and the highest scores were measured on the 14th day. With the same fermentation time, the R5 group achieved the highest scores among the four groups. The highest scores obtained by the R8 group were lower than those obtained by other groups.

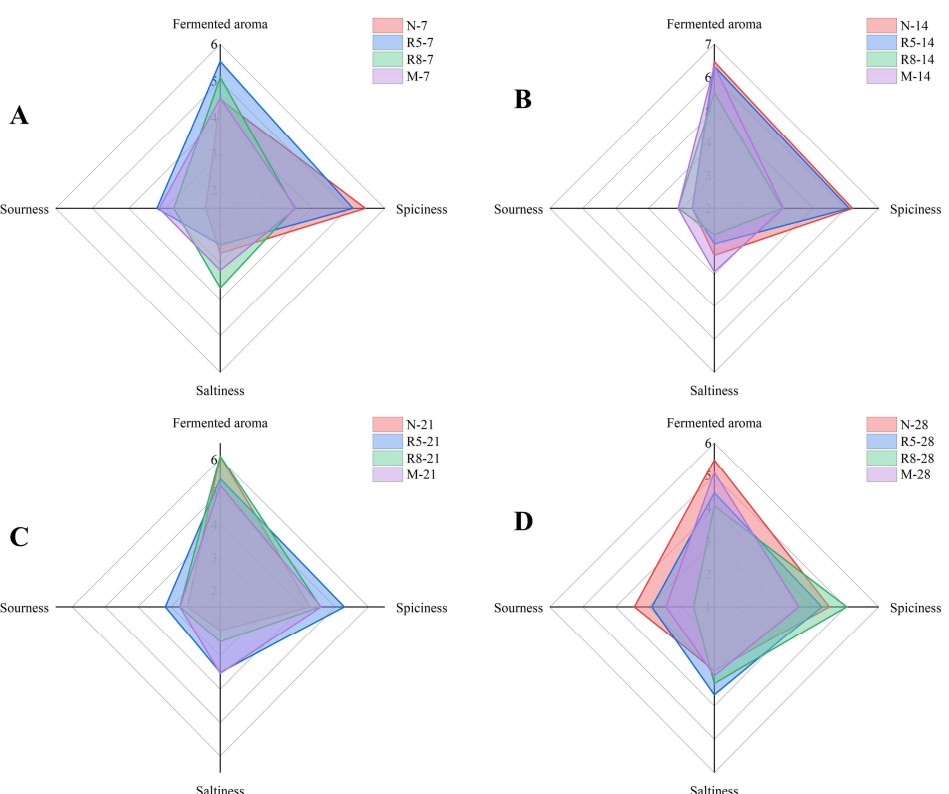

**Figure 10.** Comparison of radar charts for sensory evaluation data of pickles with different fermented times. (**A**–**D**) Pickle samples fermented for 7, 14, 21, and 28 days, respectively.

## 4. Discussion

In this study, the changes in chemical compounds, including total acid, reducing sugar, nitrite, organic acid, and VOCs, as well as pH and the bacteria community, were explored during the fermentation process using different starters (without starter, *L. plantarum* R5, *L. pentosus* R8, and R5 plus R8). The correlation between the microbial communities and chemical compounds and between the microbial communities and VOCs in different treatments were further analyzed using statistical methods and bioinformatics tools.

The pH value is a vital factor affecting the fermentation process of pickles; it could affect the growth rates of different bacteria, such as inhibiting the growth of spoilage organisms to ensure the safety of pickles [2]. In this study, the highest rate of pH decrease was observed in the R5 group, indicating that *L. plantarum* R5 can rapidly reduce the pH during fermentation. Compared with the naturally fermented group, the *Lactobacillus*-fermented group has a lower pH, meaning that microbial growth, especially *Lactobacillus*, in the *Lactobacillus*-fermented group, is different from that in the naturally fermented group. Meanwhile, the contents of total acid were increased in all groups, and these results confirmed the changes in pH in pickles.

As we all know, *Lactobacillus* can utilize carbohydrates to produce organic acids via fermentation [21]. Thus, these organic acids are essential products [22]. Moreover, numerous microorganisms could catabolize these acids as carbon sources. In addition, organic acids are essential components of the flavor in pickles, and their composition and level reflect the microbial community of pickles. Different types of organic acids were produced during fermentation (Figure 2). Lactic acid was the predominant organic acid in pickles in all samples, but the concentration of citric acid was low. In this study, the tartaric acid content in the R5 and M groups first increased, which may be due to a pH decrease, and then decreased because of oxidative degradation and tartrate precipitation [23,24]. Consistent with the results from previous reports, the content of citric acid was the lowest

among the measured organic acids, and the content of acetic acid in the different groups remained relatively low and stable [25].

In vegetables, polysaccharides, proteins, and other macromolecules can be degraded by microorganisms, especially *Lactobacillus* which gradually increases the contents of reducing sugar in pickles [26]. Moreover, with further fermentation and the continuous reproduction of microorganisms, the reducing sugar will be utilized by microorganisms or dissolved into pickle juice via osmosis and soaking. Thus, the content of the reducing sugar in all groups decreased with time (Figure 3).

The nitrite level is related to pickles' safety; a lower concentration of nitrite means more safety, and this is what consumers care about [27,28]. According to the Chinese national standard, the maximum limit of nitrite is 20 mg/kg in pickles (GB2762-2017, 2017). In our study, nitrite had the maximum concentration in the N group with 7 day fermentation, and its maximum concentration is still lower than the national standard. Meanwhile, the addition of *Lactobacillus* decreased the maximum content of nitrite, indicating *Lactobacillus* can degrade nitrites. This may be attributed to the inhibition of the nitrate-reducing bacteria, and the formation of an acidic environment with a lower pH (higher acidity) due to *L. plantarum* R5 and *L. pentose* R8.

GC-IMS was used to measure the VOCs of pickles fermented for 14 days in this study. The volatile compounds are considered one of the most critical characteristics of foods to directly affect the food acceptability for consumers. Sulfides, known as the major volatile components of fermented foods [29,30], were higher in naturally fermented samples than those in *Lactobacillus*-fermented samples. The result is consistent with the result from previous reports [31]. In our study, dimethyl disulfide, allyl methyl trisulfide, 1-Propenyl disulfide, and methyl-1-Propenyl disulfide were measured in naturally fermented samples. The sulfur compounds can be produced from cabbage, radish, red pepper, garlic, green onion, and ginger [32,33]. For example, ally alcohol and cysteine in allicin can decompose easily in garlic and generate sulfide compounds, such as methyl sulfide [34]. Esters are another important aroma compound in fermented foods with fruity odors, made from the esterification of free fatty acids with alcohol. In our study, esters in the samples inoculated with *Lactobacillus*, such as ethyl acetate, methyl acetate, and n-propyl acetate, were more plentiful than that in the spontaneously fermented samples. These esters are formed with the participation of microorganisms during lactic acid fermentation and ethanol fermentation. They are related to the *Lactobacillus* (Figure 9B). Furthermore, isobornyl acetate has been detected in Zhacai, fermented vegetables, and fresh radishes [25,35]. Butyl acetate in the R5 group showed the highest level.

Additionally, other volatile compounds, including seven alcohols, four aldehydes, four ketones, eight alkenes, two acids, one phenolic compound, and one ether compound, were also identified (Figures 5 and 6). The alcohols endowed fermented vegetables with a light, mellow flavor in pickles, and this may be related to microbial fermentation, such as *Leuconostoc, Lactobacillus,* and *Weissella* [36]. Moreover, alcohols can react with organic acids to produce esters. Aldehydes and acetone are known to be produced by hetero-fermentative bacteria, but the levels usually decrease during fermentation due to ethanol conversion by microorganisms. Therefore, aldehydes and acetone contribute little to the aroma of pickles [32]. The types and concentrations of VOCs in the samples inoculated with *Lactobacillus* were different from those in the naturally fermented pickles. Even though similar volatile compounds can be produced after *Lactobacillus* treatment, there were still different compounds produced, including cis-ocimene, isoamyl acetate, iso-propyl propanoate, 2-ethythiophene, and α-pinene oxide. The results indicated that microorganisms might be the critical factor for forming volatile compounds in fermented vegetables, and volatile compounds could vary with starters. In addition, there were significant differences in the volatile components of pickles at different fermentation stages.

There is a close relationship between microorganisms and flavor formation during fermentation. The flavor in pickles mainly comes from the metabolic activities and growth of microbial flora.

In fermented food, the flavor during fermentation is related to the microbial community, and the interaction between the material environment (raw materials and auxiliary materials, salt, temperature, etc.) and the abiotic environment that affects microorganisms. The interaction begins with the starter microorganisms existing in vegetables and other raw materials, and with the progress of fermentation, microorganisms continuously evolve. A series of metabolites produced during microbial succession ultimately gives pickles their characteristic flavor, especially the bacteria succession.

Alpha diversity showed that the diversity of the bacterial community gradually decreased, and decreased to the lowest value after the addition of *Lactobacillus* because *Lactobacillus* was the dominant strain. This result is consistent with the change in the abundance at the genus level (Figure 6B). Firmicutes and Proteobacteria are the primary phyla in the different studies on pickles, and this is in accordance with previous research [31,37,38]. In this study, *Lactobacillus* was observed as the most dominant genera during fermentation, and this was consistent with the result from most available studies on fermented vegetables [37,39–41]. *Lactobacillus* could produce lactic acid; it had a significantly positive correlation with lactic acid (Figure 8). The flavor in pickles comes mainly from the fermentation of lactic acid bacteria, which can give pickles a pleasant flavor (fruity and floral) (Figure 9B). Compared with other studies, the abundance of *Lactobacillus* in the R5 group, R8 group, and M group was much higher, and the reason for this may be due to the inoculation of *L. plantarum* R5 and *L. pentose* R8.

*Unclassified_Enterobacteriaceae*, *Pantoea*, *Weissella* and *Leuconostoc* were identified as the dominant genera in pickles [38]. They were also detected in the R5 group, R8 group, and M group, but with the progress of fermentation, the abundance decreased gradually to a relatively low level. Furthermore, our results indicated that after the R5 or R8 treatment, the overall microbial structure of pickles was not significantly affected but the microbial diversity of the naturally fermented group was increased.

The correlation among the major genera was determined in this study. According to previous studies belonging to the same genus, different species still have opposite correlations with other species. In the present study, at the genus level, *Lactobacillus* had a strong negative correlation with *unclassified Enterobacteriaceae*, *Pantoea*, and *Leuconostoc*, and this result is in accordance with previous reports.

Furthermore, the relationship between different microorganisms and chemical compounds in the sample has been visually displayed through the heatmap. Lactic acid was similar to malic acid and had a strong positive correlation with *Lactobacillus* and a negative correlation with *Streptophyta* and *Weissella*. In the later stage of fermentation, the lactic acid and malic acid provide a sour taste and react with aldehydes, ketones, and alcohols to produce a variety of new flavor substances. The esters had a strong positive correlation with *Lactobacillus*. These results were consistent.

## 5. Conclusions

This study revealed fermented pickles' chemical characteristics and bacterial diversity with different starters (without, R5, R8, and R5 plus R8) during the fermentation process. The changes in pH values suggested that the rate of *Lactobacillus* fermentation was the highest and the rate of natural fermentation was the lowest. Moreover, *Lactobacillus* enhanced the increase in total acid content, degradation of nitrite, and production of the fermented pickles' organic acid (lactic acid and malic acid). A total of 80 flavors were detected in pickles fermented for 14 days, and esters in the pickles fermented by *Lactobacillus* were more plentiful than those in the naturally fermented pickles. *Lactobacillus* was the absolute dominant genus in all groups. The correlation between these major genera was also analyzed. Comparative analysis based on non-metric multidimensional scaling (NMDS) showed that the microbial community in the pickles was affected by *Lactobacillus*.

Furthermore, the correlations between the microbial community and chemical composition were explored based on multivariate statistical analysis. Correlation analysis further showed that *Lactobacillus* correlated strongly negatively with pH, and strongly positively

with malic acid and lactic acid and the microorganisms in pickles could acclimate to the changing fermentation environment.

Overall, naturally fermented pickles showed richer VOCs and microbial diversity, but the pickles fermented by *L. plantarum* R5 only showed a higher fermentation rate than natural fermentation and the highest sensory scores. Thus, *L. plantarum* R5 can be used for the industrialized production of pickles. The insights gained from this study provide new insights into the microbiota succession and chemical compounds involved in the pickles fermented by *Lactobacillus*.

**Author Contributions:** Conceptualization, X.L. and Y.L.; methodology, S.B.; software, X.T.; validation, L.L., X.L. and X.T.; formal analysis, Y.L.; investigation, Y.L.; writing—original draft preparation, X.L.; writing—review and editing, Y.L.; supervision, Y.L.; project administration, Y.L.; funding acquisition, Y.L. All authors have read and agreed to the published version of the manuscript.

**Funding:** This research was funded by the Key Research and Development Program in Shandong Province (2019YYSP026), and the High-level Talent Introduction Project of Linyi University (LYDX2018BS032).

**Institutional Review Board Statement:** Not applicable.

**Informed Consent Statement:** Not applicable.

**Data Availability Statement:** Not applicable.

**Conflicts of Interest:** The authors declare no conflict of interest. The funders had no role in the design of the study; in the collection, analyses, or interpretation of data; in the writing of the manuscript, or in the decision to publish the results.

## Appendix A

**Table A1.** The bacterial $\alpha$-diversity index of pickles during fermentation.

| Sample | Shannon | Chao | Ace | Simpson |
|---|---|---|---|---|
| N—1 day | 2.03 | 90.00 | 94.65 | 0.19 |
| N—7 day | 1.51 | 114.22 | 160.94 | 0.38 |
| N—14 day | 1.71 | 129.35 | 133.62 | 0.30 |
| N—21 day | 1.73 | 110.55 | 113.90 | 0.29 |
| N—28 day | 1.60 | 100.05 | 103.07 | 0.35 |
| R5—1 day | 1.04 | 125.06 | 130.06 | 0.51 |
| R5—7 day | 0.61 | 127.47 | 138.82 | 0.77 |
| R5—14 day | 0.80 | 131.71 | 126.28 | 0.68 |
| R5—21 day | 0.31 | 116.20 | 119.99 | 0.91 |
| R5—28 day | 0.48 | 107.50 | 112.36 | 0.84 |
| R8—1 day | 0.62 | 99.50 | 106.27 | 0.78 |
| R8—7 day | 0.48 | 106.00 | 110.77 | 0.84 |
| R8—14 day | 0.44 | 103.25 | 109.60 | 0.86 |
| R8—21 day | 0.43 | 118.55 | 123.05 | 0.86 |
| R8—28 day | 0.75 | 109.89 | 123.66 | 0.70 |
| M—1 day | 0.22 | 107.65 | 109.52 | 0.94 |
| M—7 day | 0.70 | 113.11 | 117.22 | 0.74 |
| M—14 day | 0.42 | 112.07 | 115.03 | 0.86 |
| M—21 day | 0.43 | 125.05 | 138.59 | 0.83 |
| M—28 day | 0.29 | 137.58 | 164.31 | 0.91 |

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
