# Peer review of "Microbiota Succession and Chemical Composition Involved in Lactic Acid Bacteria-Fermented Pickles"

_fermentation, doi:10.3390/fermentation9040330_

Round 1

Reviewer 1 Report

The paper focus on the changes of chemical compounds, including total acid, reducing sugar, nitrite, organic acid and VOCs, as well as pH and the microbiota succession in the Lactic Acid Bacteria Fermented Pickles.

However, this paper slightly lacks of innovation and research highlights.

The amount of experimental data is large, and the internal correlation between microbiota succession and Chemical Composition should be further explored.

This article also has the following minor problems:

1、         Revise the fig 5.6 format to make the image clearer.

2、         The VOCs of the pickle (14 days) were determinedWhy not 21days, 28 days

Reviewer 2 Report

Dear authors

The submitted study deals with interesting theme. It provides many interesting results.

However, the submitted manuscript has several limitations, and its minor revision is needed.

Comments

 Line 151: “the pH of D” – please check this designation.

Line 173: “(249.53 and 251.34, respectively)” – please add the unit.

Line 174: “214.82 mg/g” – is the unit correct?

Page 5 of 16, Figure 3 - please check the designation “reduced sugar”. The content of reducing sugar is lower in N group than groups R5, R8, M. Could you explain it?

Line 185: “Nitrite concentration in pickle went up first, but then it went down.” Please explain what caused an increase in nitrites.

Check the unit mg/kg (Figure 4) vs. mg/g (text line 187-189).

Line 188: “… 10.72, 9.07, 9.82 mg/g, respectively.” – please add the information about what value corresponds to what sample.

The Latin names must be written in italics (check the whole manuscript).

Line 192: “compound” – correct to “compounds”.

In the section “5. Conclusions”, some sentences from the results are repeated (e.g. line 436-438).
